# CLEFT-Q SwePsych protocol: A prospective observational study to investigate the psychometric characteristics test-retest reliability, responsiveness, and interpretability of CLEFT-Q

Mia Stiernman[1,2☯*], Kristina Klintö[3,4☯], Anna-Paulina Wiedel[2,5☯],
Malin Schaar-Johansson[3,4☯], Måns Cornefjord[1,2☯], Björn Schönmeyr[2,6‡],
Gudrun Stålhand[7,8‡], Mikael Korduner[9,10‡], Martin Bengtsson[9,10‡], Maria Mani[11,12‡],
Malin Hakelius[11,12‡], Roshan Peroz[11‡], Johan Zötterman[13,14‡], Jenny Cajander[15,16‡],
Mats Sjöström[17,18‡], Justin Weinfeld,[19‡] Christina Persson[19‡], Conrad Harrison[20‡],
Alexander Allori[21‡], Anna Miroshnychenko[22‡], Magnus Becker[1,2☯]

1 Department of Plastic and Reconstructive Surgery, Skåne University Hospital, Malmö, Sweden,
2 Division of Surgery, Department of Clinical Sciences in Malmö, Lund University, Lund, Sweden,
3 Division of Speech Language Pathology, Phoniatrics and Audiology, Department of Clinical Sciences in Lund, Lund University, Lund, Sweden, 4 Department of Speech Language Pathology, Department of Otorhinolaryngology, Skåne University Hospital, Malmö, Sweden, 5 Department of Oral and Maxillofacial Surgery, Skåne University Hospital, Malmö, Sweden, 6 Plastic- and Craniofacial Surgery, Karolinska University Hospital, Stockholm, Sweden, 7 Maxillofacial Unit in Linköping, Linköping, Sweden, 8 Department of Biomedical and Clinical Sciences, Linköping University, Linköping, Sweden, 9 Department of Oral and Maxillofacial Surgery, Skåne University Hospital, Lund, Sweden, 10 Department of Clinical Sciences, Faculty of Medicine, Lund University, Lund, Sweden, 11 Department of Surgical Sciences, Uppsala University, Uppsala, Sweden, 12 Department of Plastic Surgery and Maxillofacial Surgery, Uppsala University Hospital, Uppsala, Sweden, 13 Department of Clinical and Experimental Medicine, Linköping University, Linköping, Sweden, 14 Department of Hand and Plastic Surgery and Burns, Linköping University Hospital, Linköping, Sweden, 15 Division of Hand- and Plastic Surgery, Umeå University Hospital, Umeå, Sweden, 16 Division of Surgery, Department of Surgical and Perioperative Science in Umeå, Norrland University Hospital, Umeå, Sweden, 17 Oral and Maxillofacial Surgery, Umeå University Hospital, Umeå, Sweden, 18 Department of Odontology, Umeå University, Umeå, Sweden, 19 Institute of Neuroscience and Physiology, Speech and Language Pathology Unit, Sahlgrenska Academy, University of Gothenburg, Gothenburg, Sweden, 20 Nuffield Department of Orthopaedics, Rheumatology and Musculoskeletal Sciences, University of Oxford, Oxford, United Kingdom, 21 Division of Plastic, Maxillofacial, and Oral Surgery, Duke University, Durham, North Carolina, United States of America, 22 Department of Health Research Methods, Evidence and Impact, McMaster University, Hamilton, Ontario, Canada

☯ These authors contributed equally to this work.
‡ BS,GS,MK,MB,MM,MH,RP,JZ,JC,MS,JW,CP,CH,AA and AM also contributed equally to this work.
* mia.stiernman@med.lu.se

## Abstract

### Objectives

Patient perceived benefit of treatment for cleft lip and/or palate is of great importance since it is central to development of cleft care. CLEFT-Q is a cleft-specific questionnaire on health-related quality of life. Test-retest reliability, aspects of responsiveness

**Data availability statement:** No datasets were generated or analysed during the current study. All relevant data from this study will be made available upon study completion.

**Funding:** This work was supported by The Foundation for Plastic Surgery Research, Malmö, Sweden (MS) as well as regional research funds from Region Skåne (USVE 2023-068 to MS).

**Competing interests:** The authors have declared that no competing interests exist.

and interpretability are yet to be established for CLEFT-Q. This study aims to investigate these psychometric characteristics of CLEFT-Q.

## Methods

To establish the test-retest reliability of CLEFT-Q, data will be collected repeatedly and independently at approximately 1-week intervals. Inclusion of approximately 50 patients is considered adequate for a test-retest study. To improve the interpretability of CLEFT-Q norm data from a control population of volunteers without a cleft will be collected. A total of approximately 210 participants will be included from schools, high-schools and universities. To test the responsiveness of CLEFT-Q, patients will answer selected subscales of CLEFT-Q, longitudinal anchor questions and perform global ratings of change before and after surgery. To ensure robust results, approximately 50 patients for each type of treatment will be recruited. If CLEFT-Q is found to be responsive, the pre- and postoperative difference in scores of CLEFT-Q will be compared with the change in objective measurements based on assessments by professionals in cleft care obtained in this study. To evaluate interpretability, results will be analysed to investigate the minimal important change using anchor-based, distribution-based and qualitative approach.

### Registration details

This study is registered at ClinicalTrials.gov under the ID 2021-06993-01.

### Introduction

Cleft lip and/or palate (CL/P) is one of the most common congenital anomalies. Primary surgery is performed to repair the lip, palate and alveolus. Secondary treatment includes surgery to improve speech, decrease visibility of scars, increase fullness and symmetry of the upper lip and nose, increase nose tip projection, normalize skeletal relations in the facial skeleton and improve occlusion.

Various aspects of health-related quality of life (HRQOL) can be affected by CL/P and its treatment [1,2]. Consequently, many interventions for patients with CL/P focus on increasing the quality of life and patient self-perception. Patient perspective therefor is central to evaluating results of treatment [3–6]. Patient reported outcome measures (PROMs) measure different aspects of HRQOL and accordingly provide valuable information to overall outcomes measurement and quality improvement.

CLEFT-Q™ is a CL/P-specific HRQOL questionnaire developed through a process involving review of previous field surveys [1], qualitative research on patients with CL/P from different countries [7,8], field studies in 12 different countries, and finally, analysis with Rasch measurement theory (RMT) [9]. CLEFT-Q has been translated into Swedish [10]. Normative scores for the CLEFT-Q subscales have been developed for different ages and diagnostic groups based on 2434 individuals with CL/P aged 8–29 years from 12 countries [9]. Construct validity has been appraised for

CLEFT-Q by comparing it with the PROMs Cleft Hearing Appearance and Speech Questionnaire (CHASQ) and Child Oral Health Impact Profile (COHIP) [11]. Several CLEFT-Q subscales were selected for inclusion in the standard set of outcomes for cleft care recommended by the International Consortium for Health Outcomes Measurement (ICHOM) [12]. Continuous effort is being made to refine the standard set and minimising the burden on patients by selecting the most meaningful scales for different subpopulations [13,14] and by developing computerised adapting testing for CLEFT-Q [15].

Previous publications have discussed remaining methodological problems within PROMs in the CL/P population [2,16]. Some were, for example, the lack of qualitative research as well as the lack of studies following the development of patient satisfaction longitudinally and change of satisfaction from before to after interventions. It is also important to describe the level of health within the general population. This facilitates the identification of a cut-off level for pathological values in the CL/P population and when further examination or intervention is indicated. Therefore, including a control population in studies on psychosocial health in individuals with CL/P has also been strongly recommended [2,16–18].

Several areas of interest regarding reliability and validity of CLEFT-Q are still unexplored. These include test-retest reliability, responsiveness and interpretability. Test-retest reliability is defined by the COSMIN initiative (COnsesus based Standards for the selection of health Measurement Instruments) as 'The extent to which a score for patients who have not changed are the same for repeated measurement over time' [19]. Further, the COSMIN initiative defines responsiveness as 'The ability of a PROM to detect change over time in the construct to be measured' [19]. Testing this characteristic of a PROM is crucial to know if it is suitable for longitudinal follow-up and detecting change before and after treatment. Finally, the COSMIN initiative defines interpretability as 'The degree to which one can assign qualitative meaning – that is, clinical or commonly understood connotations – to an instrument's quantitative scores or change in scores' [19]. In investigating the interpretability of a PROM, one may examine the scores in a control population as well as the minimal clinical important change (MIC) [20].

With these concepts in mind, there remains a critical need to more thoroughly evaluate the CLEFT-Q scales. Specifically, this project will appraise the following:

1. Test-retest reliability of CLEFT-Q

2. Responsiveness of CLEFT-Q

3. MIC of CLEFT-Q

4. Control population values for CLEFT-Q

## Materials and methods

This project includes several studies which address four separate aims related to the validity and reliability of CLEFT-Q. The methods and analysis for each study will be described separately. A SPIRIT Checklist for this protocol is available as Supporting Information. The Swedish Ethical Review Authority approved this project on 2022-02-15 under the ID nr 2021-06993-01. Major amendments to the trial protocol will be approved by the ethical review authority before being carried out. Local healthcare professionals will inform participants verbally, provide written information, and collect signed consent forms before inclusion in the project. Study data will be collected and managed using REDCap electronic data capture tools hosted at Lund University [21,22]. Results will be disseminated via publications in open access scientific journals and at national and international congresses. Recruitment and data collection began in 2022-09-17. Participant recruitment and data collection is predicted to be completed in 2027 and results are expected in 2028.

### Participants

The common inclusion criteria for all studies in the project CLEFT-Q SwePsych are participants between 8 and 29 years of age who can understand written or spoken Swedish. This age interval aligns with earlier validation of CLEFT-Q [23].

Sufficient knowledge of Swedish is crucial for inclusion in the studies since participants must be able to answer the PROM independently from the health care provider. Exclusion criteria are age < 8 or > 29 and insufficient knowledge of Swedish. Patients lost to follow-up will be analysed regarding cleft subtype, sex, age, intervention, and additional diagnoses.

**Study 1: Test-retest reliability.** Quantitative data from CLEFT-Q will be collected repeatedly and independently at approximately 1-week intervals. In the period between the two measurements, no interventions will take place. Results from the two different measurement points will be compared to see how reliably CLEFT-Q measures HRQOL. The intraclass correlation coefficient (ICC) will be calculated. Since CLEFT-Q scores are continuous data, analysis of variance (ANOVA) models will be used to calculate test-retest reliability. Inclusion of approximately 50 patients is considered adequate for a test-retest study [20,24]. According to a formula for calculating sample size in the study of de Vet et al. (2011), fifty patients are required to reach an ICC of 0.8 with a confidence interval of 95% +/- 0.1 in a design with two repeated measurements. Patients will be recruited consecutively until 50 patients have been included. They will be recruited at routine check ups at 10, 13, 16 and 19 years of age. Results will be presented according to guidelines for test-retest reliability [25]. Smallest Detectable Change (SDC) will also be calculated [20].

**Study 2: Responsiveness.** Participants will be recruited from all six CL/P centres in Sweden – Sahlgrenska University Hospital in Gothenburg, Karolinska University Hospital in Stockholm, Linköping University Hospital, Skåne University Hospital in Malmö, Umeå University Hospital and Uppsala University Hospital. Patients will answer selected subscales of CLEFT-Q and global ratings of change (GRCs) before and after secondary CL/P surgery. The participant timeline is shown in Fig 1. Specific subscales of CLEFT-Q and clinical measurements relevant for each group of secondary CL/P surgery are listed in Table 1.

A construct approach to testing responsiveness will be carried out. All hypotheses are made a priori and listed in Table 2. Responsiveness can be considered high if > 75% of hypotheses are confirmed. A level of 50–75% confirmed hypotheses is considered moderate, and < 50% confirmed hypotheses is considered low responsiveness [20]. Hypotheses will also be ranked according to perceived importance for each type of surgery. More important hypotheses will be given

| | STUDY PERIOD | | | | |
|---|---|---|---|---|---|
| | **Enrolment** | **Pre-intervention data collection** | **Intervention** | **Follow-up** | **Close-out** |
| **TIMEPOINT** | *-t₁* | *Before surgery* | *Surgery* | *6 or 12 months* | |
| **ENROLLMENT:** | | | | | |
| Eligibility screen | X | | | | |
| Informed consent | X | | | | |
| **INTERVENTIONS** | | | | | |
| Secondary surgery | | | X | | |
| **ASSESSMENTS:** | | | | | |
| CLEFT-Q | | X | | X | |
| Clinical data | | X | | X | |
| Longitudinal anchor | | X | | X | |
| Semi-structured interview | | | | X | |
| Global rating of change | | | | X | |
| Other interventions/ major life events control question | | | | X | |
| Patient characteristics | | | | | X |

**Fig 1. Participant timeline.**

**Table 1. Specific subscales of CLEFT-Q and clinical measurements relevant for each group of secondary CL/P surgery.**

|  | Nose surgery | Lip surgery | Speech surgery | Orthognatic surgery |
|---|---|---|---|---|
| CLEFT-Q sub-scales | Face, Nose, Nostrils, Psychological function, Social function, School function | Face, Lips, Cleft lip scar, Psychological function, Social function, School function | Speech distress, Speech function, Psychological function, Social function, School function | Face, Jaw, Teeth, Psychological function, Social function, School function, Eating and drinking |
| Clinical data | Photograph | Photograph | Audio-recording | Photograph, cephalometric measurements on x-ray, dental cast data |

greater influence on the analysis of results alongside the numerical, but arbitrary, cut-offs for levels of responsiveness mentioned above. In Table 2 the hypotheses are listed in order of importance.

GRC (Global Rating of Change)

To ensure robust results, approximately 50 patients for each treatment will be recruited [24,26]. Hypotheses for the magnitude of change are based on normative data from the CLEFT-Q field test [9]. For example, on the subscale Nose, the mean score for patients with cleft lip and palate (CLP) was 53.5 points and the mean score for patients with cleft palate only (CP) was 71.7 points [9]. The difference between the two mean scores is 71.7–53.5 = 18.2 points. Patients with CP are hypothesised to have scores close to that of the general population, since their noses are not affected by their cleft. Patients who undergo a surgical intervention to improve their nose are hypothesised to improve their appearance but not erase the underlying deformity completely. They are therefore hypothesised to improve their CLEFT-Q Nose score somewhat less than 18 points. In this study, the magnitude of improvement of approximately 15 points was chosen.

Hypotheses regarding differences in magnitude of different CLEFT-Q subscales are based on how well the subscale is thought to capture the change in HRQOL due to the treatment. Correlations between changes in subscale scores, treatment-specific longitudinal anchors and GRC outcomes will also be analysed to test responsiveness. Hypotheses of differences in the strength of correlations are derived from the similarity between the constructs of the treatment-specific longitudinal anchor, GRC, and the CLEFT-Q subscale.

Calculations regarding correlation hypotheses will be carried out with Spearman's rank correlation. Correlation strength will be judged as high (≥0.5), moderate (0.3–0.5), or weak (<0.3) in accordance with earlier studies [27]. Power calculations have been carried out to ensure that sample sizes are also adequate to detect a statistical difference between pre- and postoperative scores if the hypotheses regarding magnitude are satisfied. The power calculations are based on normative data from the CLEFT-Q field test [9]. For example, in the normative data from CLEFT-Q the mean Nose subscale score for patients with CL/P was 53.5 and the standard deviation was 21.9. With alpha set at 0.05 and power set to 80%, 50 participants is an adequate sample to ensure that 15 points improvement will be statistically significant (a postoperative Nose subscale score of 53.5 + 15 = 68.5 points).

Pre- and postoperative audio recordings, photographs and cephalometric measurements will be collected. The entire population in each study will be judged by the same raters. For example, all the patients who have had secondary lip surgery will be rated by the same panel of plastic surgeons. Photographs, audio recordings and radiographs will be cropped or edited so that assessors cannot identify individuals, at which center they were treated or whether they are pre- or postintervention.

For nose and lip surgery, photographs will be cropped and rated by three plastic surgeons according to the method described by Asher-McDade et al. (1991) [28]. A change in one point out of the five point scale described by Asher-McDade et al will be used as a an indicator for successful surgery. For orthognatic surgery, dental overjet, overbite and sectioned Modified Huddart/Bodenham (MHB) index will be analysed on dental casts by a panel of three orthodontists or craniofacial surgeons. Photographs of front view, profile view and cephalometric measurements on x-rays will be included to analyse the angles between cefphalometric landmarks sella, nasion and point A (SNA), sella, nasion and point B (SNB), point A,

**Table 2. A priori hypotheses for the responsiveness of CLEFT-Q.**

| Treatment | Hypotheses |
|---|---|
| Nose surgery | • Magnitude of improvement of Nose subscale score will be approximately 15 points.<br>• Magnitude of improvement of Nostrils subscale score will be approximately 15 points.<br>• Nose and Nostrils subscale scores will correlate highly and positively with nose specific GRC.<br>• Face subscale score will correlate moderately and positively with nose specific GRC.<br>• Change in Nose subscale scores will correlate strongly and positively with change in nose specific longitudinal anchor score.<br>• Change in Nostrils subscale scores will correlate strongly and positively with change in nose specific longitudinal anchor score.<br>• Change in Face subscale scores will correlate moderately and positively with change in longitudinal anchor score.<br>• Nose and Nostril subscale scores will improve more than Face subscale score.<br>• Nose, Nostrils and Face subscale scores will improve more than psychosocial subscale scores.<br>• Weak correlations, positive or negative, are expected between psychosocial subscale scores and nose specific GRC.<br>• Weak correlations, positive or negative, are expected between psychosocial subscale scores and change in longitudinal anchor score. |
| Lip surgery | • Magnitude of improvement of Lips subscale score will be approximately 10 points.<br>• Lip subscale score will correlate highly and positively with lip specific GRC.<br>• Face subscale score will correlate moderately and positively with lip specific GRC.<br>• Change in Lip subscale scores will correlate strongly and positively with change in lip specific longitudinal anchor score.<br>• Change in Lip scar subscale scores will correlate strongly and positively with change in lip scar specific longitudinal anchor score.<br>• Change in Face subscale scores will correlate moderately and positively with change in longitudinal anchor scores.<br>• Lips subscale score will improve more than Face subscale score.<br>• Lips and Face subscale scores will improve more than psychosocial subscale scores.<br>• Weak correlations, positive or negative, are expected between psychosocial subscale scores and lip specific GRC.<br>• Weak correlations, positive or negative, are expected between psychosocial subscale scores and change in longitudinal anchor scores. |
| Orthognatic surgery | • Magnitude of improvement of Jaws subscale score will be approximately 10 points.<br>• Jaws subscale scores will correlate highly and positively with jaws specific GRC.<br>• Face subscale score will correlate moderately and positively with jaws specific GRC.<br>• Change in Jaws subscale scores will correlate strongly and positively with change in jaw specific longitudinal anchor score.<br>• Change in Teeth subscale scores will correlate strongly and positively with change in teeth specific longitudinal anchor score.<br>• Change in Face subscale scores will correlate moderately and positively with change in longitudinal anchor scores.<br>• Jaws subscale scores will improve more than Face subscale score.<br>• Jaws and Face subscale scores will improve more than psychosocial subscale scores.<br>• Weak correlations, positive or negative, are expected between psychosocial subscale scores and jaws specific GRC.<br>• Weak correlations, positive or negative, are expected between psychosocial subscale scores and change in longitudinal anchor scores. |
| Speech improving surgery | • Magnitude of improvement of Speech function subscale score will be approximately 10 points.<br>• Magnitude of improvement of Speech distress subscale score will be approximately 10 points.<br>• Speech function subscale score will correlate highly and positively with speech specific GRC.<br>• Speech distress subscale score will correlate moderately and positively with speech specific GRC.<br>• Change in Speech distress subscale scores will correlate strongly and positively with change in longitudinal anchor score.<br>• Change in Speech function subscale scores will correlate moderately and positively with change in longitudinal anchor score.<br>• Speech function and Speech distress subscale scores will improve more than psychosocial subscale scores.<br>• Weak correlations, positive or negative, are expected between psychosocial subscale scores and speech specific GRC.<br>• Weak correlations, positive or negative, are expected between psychosocial subscale scores and change in longitudinal anchor score. |

nasion and point B (ANB), and nasion, point A and pogonion (NAPg). A change of four points on the MHB will be used as an indicator for successful surgery. Speech will be documented with standardized audio recordings using the Swedish Test for Nasality and Articulation (SVANTE) [29]. The recordings will be rated independently by three blinded speech-language pathologists specialised in cleft palate speech, which is considered as gold standard [30]. Velopharyngeal competence will be rated using the VPC-Sum and the VPC-Rate [31]. Intra and inter reliability of ratings will be reported. A one-point change of the median score on either speech scale will be used as an indicator for successful surgery.

Depending on the change in the clinical outcomes described above, patients will be divided into successful and unsuccessful treatment subpopulations. The hypotheses for responsiveness of CLEFT-Q will be tested in the total population as well as in the subpopulations.

**Study 3: Minimal clinical important change.** Patients will complete selected subscales of CLEFT-Q pre- and postoperatively, see Fig 1. Results will be analysed to investigate MIC using an anchor-based approach, a distribution-based approach and a qualitative approach.

**Anchor-based approach.** Patients will be divided into subpopulations of 'changed' and 'not changed' according to their answers to the longitudinal and retrospective anchors (the GRC). The anchors will be specific to each studied intervention and subscale of CLEFT-Q. The GRC will explicitly mention the specific condition under treatment, construct under examination, time anchor points, and an ordinal scale with seven steps defining change in pre- to postoperative results [32]. An example of the nose surgery specific GRC is demonstrated in Fig 2.

Based on the GRC and longitudinal anchor, differences in CLEFT-Q scores will be analysed with a receiver operating characteristic (ROC) method to test CLEFT-Qs ability to distinguish between participants who have changed and those who haven't changed after the intervention. This study will use items and response options from CHASQ as longitudinal anchors. Sensitivity and 1- specificity will be plotted on the ROC-curve for different values of MIC. The optimal value, depending on preferred levels of sensitivity and specificity, will be determined and presented according to the visual anchor-based MIC distribution method [33]. The mean difference in PROM-scores between 'changed' and 'not changed' subpopulations will also be presented as an estimate of MIC. Patients will be divided into subpopulations of successful and unsuccessful treatment depending on clinical change (on audio recordings, photographs, x-ray and/or dental casts). The MIC calculations of CLEFT-Q will be tested in both the total population and the subpopulations.

**Distribution-based approach.** In agreement with Norman et al. 2003, a distribution based approach will be used to calculate a value of MIC as 0.5*SD of baseline data collected with CLEFT-Q [34].

**Qualitative approach.** Qualitative data will be collected through semi-structured interviews pre- and postoperatively. Interview templates specific to treatment including topics based on a literature review will guide the discussion [7,8,35–38]. Examples of topics are: What are the main expectations before surgical intervention? Have the results after surgical interventions met patient expectations? Are there concepts of importance that CLEFT-Q does not examine? The interview template may be adapted during the study depending on patient answers. Interviews will be recorded and transcribed verbatim. Data will be collected until thematic saturation is achieved (when further interviews no longer contribute to new emerging themes). Interview data will be examined with thematic analysis to find common themes in the material [39]. Themes and quotes will be checked with 10 study participants to ensure representativeness and reliability. A clear pathway of referral to a counsellor will be presented to the patient if such a need should become evident during the interview [40]

**Study 4: Control population for CLEFT-Q.** Determining scores of CLEFT-Q in a control population without CL/P is one aspect of investigating the interpretability of the PROM [20]. CLEFT-Q results will be collected from volunteers in

*How much do you like the appearance of your nose now compared to before the surgery of your nose?*

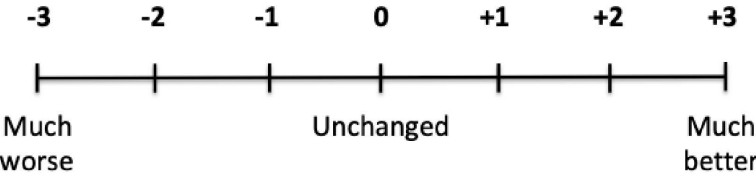

**Fig 2. Nose surgery-specific global rating of change (GRC).**

schools and high schools who were not born with a cleft. A descriptive presentation of the mean scores of the subscales in CLEFT-Q for the control population will be presented. These mean scores can then be compared to individuals and populations with CL/P.

The control population will be divided into six subpopulations (girls, boys and ages 8–11 years, 12–16 years, and 17 years and older). Approximately 35 participants per control subpopulation will be included, resulting in 210 healthy volunteers. Power calculations are based on normative data from the CLEFT-Q field test [9]. For example, on the subscale Lips, the mean score for patients with CLP was 58.2 points and the standard deviation 24.8. The mean score for patients with CP was 74.4 points and the standard deviation was 23.1 [9]. Healthy volunteers without CLP are hypothesised to have scores at least as high as patients with CP. Sample sizes are calculated to detect a difference of 74.4–58.2 = 16.2 points with alpha set at 0.05, power set to 80% and standard deviation set to 24. Sample sizes regarding speech are based on differences in scores between subpopulations of patients with CLP, and cleft lip and alveolus (CLA).

## Discussion

There is a general lack of CL/P HRQOL research to generate longitudinal data, data from before and after interventions as well as data from a control population. There is also a specific need to further evaluate the test-retest reliability, responsiveness and MIC of CLEFT-Q with a longitudinal study design. The studies presented in this protocol will specifically evaluate these aspects of reliability and validity of CLEFT-Q with both longitudinal and cross-sectional data.

Responsiveness has previously been tested for CLEFT-Q using a longitudinal study design [41]. The study included CL/P patients undergoing revision surgery of the nose or lip or orthognathic surgery. CLEFT-Q was found to be responsive. Limitations included a small sample size (18–31 patients per intervention), and that many participants were lost to follow-up (26–44% per intervention). Additionally, speech scales could not be evaluated since the study did not include speech-improving interventions [41]. Study 2 in this protocol aims to establish responsiveness with larger sample sizes and includes speech-improving interventions.

MIC has previously been tested for CLEFT-Q using two distribution-based approaches: 0.5*Standard Deviation and 0.5*Effect Size [41]. MIC estimates ranged between 5.9–14.4 points of 100. Limitations were a small sample size (18–31 patients per intervention) and that many participants were lost to follow-up (26–44% per intervention). Additionally, speech scales could not be evaluated since the study design did not include speech-improving interventions. The authors recommend that future studies include a larger population, evaluation of more subscales of CLEFT-Q, and include an anchor-based approach. Study 3 in this protocol will complement current research findings on MIC with larger sample sizes, an anchor-based approach and by including speech-improving interventions.

In summary, patient perceived benefit of treatment is of great importance since it is central to care development. The use of a high quality PROM to capture the patient perspective is a fundament of this type of research. This study will add to the examination of the quality of CLEFT-Q by assessing test-retest reliability, responsiveness and interpretability. Results from this study will immediately translate into improved interpretation of patient reported outcomes and inform treatment of patients with CL/P.

## Supporting information

**S1 File. CLEFT-Q SwePsych protocol SPIRIT Checklist.**
(DOC)

## Author contributions

**Conceptualization:** Mia Stiernman, Kristina Klintö, Anna-Paulina Wiedel, Malin Schaar-Johansson, Måns Cornefjord, Magnus Becker.

**Investigation:** Mikael Korduner, Martin Bengtsson, Björn Schönmeyr, Johan Zötterman, Gudrun Stålhand, Maria Mani, Malin Hakelius, Roshan Peroz, Jenny Cajander, Mats Sjöström, Justin Weinfeld, Christina Persson.

**Methodology:** Mia Stiernman, Kristina Klintö, Anna-Paulina Wiedel, Malin Schaar-Johansson, Måns Cornefjord, Conrad Harrison, Alexander Allori, Anna Miroshnychenko, Magnus Becker.

**Project administration:** Mia Stiernman.

**Resources:** Mia Stiernman.

**Writing – original draft:** Mia Stiernman.

**Writing – review & editing:** Kristina Klintö, Anna-Paulina Wiedel, Malin Schaar-Johansson, Måns Cornefjord, Mikael Korduner, Martin Bengtsson, Björn Schönmeyr, Johan Zötterman, Gudrun Stålhand, Maria Mani, Malin Hakelius, Roshan Peroz, Jenny Cajander, Mats Sjöström, Justin Weinfeld, Christina Persson, Conrad Harrison, Alexander Allori, Anna Miroshnychenko, Magnus Becker.

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
