## [Decision Letter · Decision Letter 0]

21 Jan 2025

PONE-D-24-49310CLEFT-Q SwePsych Protocol: a prospective observational study to investigate the psychometric characteristics test-retest reliability, responsiveness, and interpretability of CLEFT-QPLOS ONE

Dear Dr. Stiernman,

Thank you for submitting your manuscript to PLOS ONE. After careful consideration, we feel that it has merit but does not fully meet PLOS ONE’s publication criteria as it currently stands. Therefore, we invite you to submit a revised version of the manuscript that addresses the points raised during the review process.

There are some minor clarifications that require the authors attention.

We look forward to receiving your revised manuscript.

Kind regards,

James J Cray Jr., Ph.D.

Academic Editor

PLOS ONE

“Regional research funds from Region Skåne (2023-068).”

3. Please include your tables as part of your main manuscript and remove the individual files. Please note that supplementary tables (should remain/ be uploaded) as separate "supporting information" files

Reviewers' comments:

Reviewer's Responses to Questions

**Comments to the Author**

1. Does the manuscript provide a valid rationale for the proposed study, with clearly identified and justified research questions?

Reviewer #1: Yes

2. Is the protocol technically sound and planned in a manner that will lead to a meaningful outcome and allow testing the stated hypotheses?

Reviewer #1: Yes

3. Is the methodology feasible and described in sufficient detail to allow the work to be replicable?

Reviewer #1: No

4. Have the authors described where all data underlying the findings will be made available when the study is complete?

Reviewer #1: Yes

5. Is the manuscript presented in an intelligible fashion and written in standard English?

Reviewer #1: Yes

6. Review Comments to the Author

You may also provide optional suggestions and comments to authors that they might find helpful in planning their study.

Reviewer #1: The authors describe the methodology for a series of studies designed to provide the test-retest reliability, responsiveness, MIC and control population values for CLEFT-Q. I commend the authors for undertaking this important work, the results of which will benefit future researchers and clinicians alike. I recommend this manuscript to be accepted for publication with minor clarifications, listed below.

Study 1:

Data collection is stopped at 50 participants. How is a sufficient representation of patients across the questionnaire age range ensured?

Study 2:

Line 242 “Power calculations have been carried out” – Where/how?

Line 265 abbreviations should be spelled out for the angles

How were the numerical hypotheses for the magnitude of improvement, listed in table 2, arrived at?

The study is carried out at several hospitals. Who will the assessment of change in the objective and subjective outcome measures be done by? Same people for all of the patients or different ones for each location? Will the appearance be rated by a lay panel or physicians, and what scale will be used? How many assessors will be used for each parameter? What are the criteria for successful and unsuccessful operations?

Study 3 OK

Study 4:

With the participants recruited from schools and high schools, how will the upper range of the questionnaire target group (eight to 29 years old) be sufficiently represented?

I recommend that the manuscript undergoes language editing.

7. PLOS authors have the option to publish the peer review history of their article (what does this mean? ). If published, this will include your full peer review and any attached files.

**Do you want your identity to be public for this peer review?** For information about this choice, including consent withdrawal, please see our Privacy Policy .

Reviewer #1: No

---

## [Author Response · Author response to Decision Letter 1]

25 Feb 2025

Response to Reviewers

Thank you for your effort, comments and clearly described points for improvement. We feel that the changes have improved the manuscript. Below are responses to each point raised by the academic editor and reviewer.

Manuscript has been edited to meet PLOS ONE’s style requirements.

2. Please state what role the funders took in the study.

Funding statement has been added to the revised Cover Letter.

3. Please include your tables as part of your main manuscript and remove the individual files.

Tables have been included into the main manuscript.

4. Please review your reference list

The reference list has been reviewed and updated with the following references:

Mokkink LB, Terwee CB, Patrick DL, Alonso J, Stratford PW, Knol DL, et al.

The COSMIN checklist for assessing the methodological quality of studies on measurement properties of health status measurement instruments: an international Delphi study. Qual Life Res. 2010;19(4):539-49.

Lohmander A, Hagberg E, Persson C, Willadsen E, Lundeborg I, Davies J, et al. Validity of auditory perceptual assessment of velopharyngeal function and dysfunction - the VPC-Sum and the VPC-Rate. Clinical linguistics & phonetics. 2017;31(7-9):589-97.

Suggestions and comments from Reviewer 1

Study 1:

Data collection is stopped at 50 participants. How is a sufficient representation of patients across the questionnaire age range ensured?

Participants will be recruited at routine check ups at 10, 13, 16 and 19 years of age. This way the study population will represent the most relevant age span. The CLEFT-Q will be used most often within this age span, both at routine check ups, and as part of the work up for secondary interventions.

Study 2:

Line 242 “Power calculations have been carried out” – Where/how?

Power calculation is now described with an example in the manuscript.

Line 265 abbreviations should be spelled out for the angles

Abbreviations have been spelled out in the manuscript.

How were the numerical hypotheses for the magnitude of improvement, listed in table 2, arrived at?

They are based on the normative values from the CLEFT-Q field test. An example has been added to the manuscript.

The study is carried out at several hospitals. Who will the assessment of change in the objective and subjective outcome measures be done by? Same people for all of the patients or different ones for each location? Will the appearance be rated by a lay panel or physicians, and what scale will be used? How many assessors will be used for each parameter? What are the criteria for successful and unsuccessful operations?

The following clarifications have been made in the manuscript:

• The assessment of change in clinical outcomes for speech, nose, lips and jaw surgery will be carried out by panels of professionals.

• Three plastic surgeons will rate photographs, three SLPs will rate audio recordings and three orthodontists or craniofacial surgeons will rate the cefalometric measurements.

• The entire population in each study will be judged by the same raters. For example, all the patients who have secondary lip surgery will be rated by the same panel of plastic surgeons. Photographs, audio recordings and radiographs will be cropped or edited so that assessors cannot identify individuals, which center they were treated at or whether they are pre- or postintervention

• A change of 1 point on the Asher-McDade 5 point scale will be used as a cut off for successful operation.

• A change of four points on the MHB will be used as an indicator for successful surgery.

• Velopharyngeal competence will be rated using the VPC-Sum and the VPC-Rate. A one-point change of the median score on either speech scale will be used as an indicator for successful surgery.

Study 4:

With the participants recruited from schools and high schools, how will the upper range of the questionnaire target group (eight to 29 years old) be sufficiently represented?

Participants will be recruited from schools, high schools and universities to ensure representation from the upper range of the target group. This has been clarified in the manuscript.

I recommend that the manuscript undergoes language editing.

Language editing has been carried out.

---

## [Editor Report · Decision Letter 1]

28 Feb 2025

CLEFT-Q SwePsych Protocol: a prospective observational study to investigate the psychometric characteristics test-retest reliability, responsiveness, and interpretability of CLEFT-Q

PONE-D-24-49310R1

Dear Dr. Stiernman,

We’re pleased to inform you that your manuscript has been judged scientifically suitable for publication and will be formally accepted for publication once it meets all outstanding technical requirements.

Kind regards,

James J Cray Jr., Ph.D.

Academic Editor

PLOS ONE
---

## [Editor Report · Acceptance letter]

PONE-D-24-49310R1

PLOS ONE

Dear Dr. Stiernman,

I'm pleased to inform you that your manuscript has been deemed suitable for publication in PLOS ONE. Congratulations! Your manuscript is now being handed over to our production team.

Kind regards,

on behalf of

Dr. James J Cray Jr.

Academic Editor

PLOS ONE